# Zinc and Strontium-Substituted Bioactive Glass Nanoparticle/Alginate Composites Scaffold for Bone Regeneration

**DOI:** 10.3390/ijms24076150

**Published:** 2023-03-24

**Authors:** Parichart Naruphontjirakul, Piyaphong Panpisut, Somying Patntirapong

**Affiliations:** 1Biological Engineering Program, Faculty of Engineering, King Mongkut’s University of Technology Thonburi, 126 Pracha Uthit Rd., Bang Mod, Thung Khru, Bangkok 10140, Thailand; 2Faculty of Dentistry, Thammasat University, Pathum Thani 12120, Thailand; 3Research Unit in Dental and Bone Substitute Biomaterials, Thammasat University, Pathum Thani 12120, Thailand

**Keywords:** alginate, bioactive glass nanoparticle, scaffold, strontium, zinc

## Abstract

The global population is growing older and entering an aging society. Aging results in severe tissue disorder and organ dysfunction. Bone-related injuries are particularly significant. The need for alternative bone replacement materials for human implants has grown over the past few decades. Alginate has the potential for use as a cell scaffold for bone tissue engineering due to its high bio-compatibility. To improve the bioactivity of alginate scaffolds, zinc- and strontium-containing sol-gel-derived bioactive glass nanoparticles (Zn-Sr-BGNPs) with sizes ranging from 100 to l40 nm were incorporated. Zn-Sr-BGNPs synthesized through the sol-gel process have a high sur-face-to-volume ratio, homogeneity, and purity, resulting in faster degradation. The therapeutic bivalent ions released from Zn-Sr-BGNPs strengthen the cell scaffold and improve the stimulation of the production and development of bone cells. Zn-Sr-BGNPs with different Zn to Si nominal ratios of 0, 1, and 1.5 were mixed with alginate in this research. The ratio of Zn in Zn-Sr-BGNPs and the ratio of Zn-Sr-BGNPs in scaffolds impact the pore size, swelling, and biological properties of synthesized composite scaffolds. The surface area and pore volume of a 1:1 1Zn-Sr-BGNP:Alg composite scaffold were 22.58 m^2^/g and 0.055 cm^3^/g, respectively. The incorporation of Zn-Sr-BGNPs improved the mechanical performance of the scaffolds up to 4.73 ± 0.48 MPa. The swelling rate decreased slightly from 2.12 (pure Alg) to 1.50 (1Zn-Sr-BGNP:Alg (1:1)). The 1Zn-Sr-BGNP:Alg (1:1) composite scaffold promoted bioactivity through apatite layer formation, increased bone cell proliferation via the dissolution products released from the scaffold, enhanced calcium deposition, and facilitated cell attachment. Thus, 1Zn-Sr-BGNP:Alg (1:1) composite scaffold is proposed as a possible artificial bone scaffold in bone tissue regeneration.

## 1. Introduction

According to the World Health Organization (WHO), more people are entering the aging society, especially in Asia, where the elderly population increased by at least 3% annually. Ill health and unhealthy lifestyles when younger are the causes of chronic diseases in the elderly [1]. The self-remodeling ability of bone is limited in the case of severe bone damage, such as trauma, accidents, and osteoporosis, especially in the elderly. The standard treatment is to replace the damaged area with bone from the patient (autograft bone) via tissue transplantation. However, there are limitations to using autografted bones, including limited donor sites, a high risk of infection, extensive recovery times, and high cost. Therefore, a synthetic bone graft substitute has been developed for implantation in the human body [2]. Both inorganic and organic materials have been used as temporary 3-D scaffolds [3].

Alginate is a polysaccharide derived naturally from brown seaweed. The molecular network within the alginate structure consists of β-(1→4)-linked D-mannuronic acid (M) and α-(1→4)-linked L-guluronic acid (G). In the presence of certain divalent cations, such as Ca^2+^, alginate can form ionic crosslinking hydrogels through the G-blocks in its structure. Alginate has potential for use as a cell scaffold for bone tissue engineering due to its excellent biocompatibility and biodegradability [4]. When alginate contacts a physiological environment, it converts into a softer hydrogel with poor mechanical strength and stability. This transformation limits its use as a load-bearing body component. To overcome this limitation and improve bioactivity, bioactive materials, including hydroxyapatite, bioactive glass, bio-silica, collagen, and chitosan, have been introduced into the alginate crosslinking for use as a composite material [5,6,7].

Bioactive glasses (BGs) are a surface-reactive glass–ceramic biomaterial and include the original bioactive glass, which is a solid, nonporous, and hard material [8,9]. The first bioglass (Bioglass^®^ 45S5) was mainly composed of 45 wt% SiO_2_, 24.5 wt% CaO, 24.5 wt% Na_2_O, and 6.0 wt% P_2_O_5_. BGs are amorphous, and their structure is divided into network formers, network modifiers, and intermediate oxides. The network formers form glasses without additional components. The network modifiers affect the glass structure by bonding to non-bridging oxygen atoms, and the intermediate oxides act like network modifiers or network formers [10]. The unique properties of BGs are the ability to form bonds between tissues and materials by forming a hydroxyapatite layer and the capacity to release ions to induce bone formation [8]. Bioactive glass nanoparticles (BGNPs) have been developed because of their benefits over the macrostructure, including a large surface area with a higher dissolution rate and faster apatite formation. BGNPs are widely synthesized through the sol-gel technique, which hydrolyzes and condenses the alkoxide precursor to form a primary particle structure [11]. The sol-gel-derived BGNPs exhibited high purity, homogeneity, porosity, and bioactivity [12].

Therapeutic inorganic ions included in BGNPs can modify their bioactivity and specific functions, including osteogenesis, angiogenesis, antibacterial activity, and anti-cancer properties [13] because the mechanism of bone formation in the human body is related to elements such as calcium (Ca), phosphate (P), zinc (Zn), magnesium (Mg), strontium (Sr), copper (Cu), manganese (Mn), cerium (Ce) and boron (B) [14,15]. Sr and Zn have been substituted for Ca in the binary sol-gel glass system to promote bone growth [16,17,18]. BGNPs have been used as the filler in the biopolymer scaffold of tissue engineering to increase the mechanical properties, antibacterial efficiency, and bioactivity of alginate scaffolds [19]. Thus, this study used Zn and Sr-containing BGNPs (Zn-Sr-BGNPs) to modify the properties of alginate scaffolds.

BGs have been doped into alginate scaffolds to improve bioactivity due to the therapeutic ions in the BG composition that stimulated a specific molecular response [20,21]. Melt–quench-derived BGs modified with therapeutic ions, such as Sr, Zn, Mg, and Cu, have been incorporated into alginate scaffolds to improve mechanical properties and biological activity [22]. However, the incorporation of sol-gel-derived BGNPs in the alginate scaffolds has rarely been reported. BGNPs containing Zn and Mg have been loaded into alginate scaffolds to improve osteoblast differentiation [19]. The unique physicochemical properties and biological performance of sol-gel–derived BGNPs, including high specific surface-to-volume ratio, rapid biodegradability, and fast release of therapeutic ions, caused their consideration as the filler in the composite materials. Therefore, the Zn- and Sr-containing sol-gel-derived BGNPs (Zn-Sr-BGNPs) with a diameter size of 100–140 nm were incorporated in the alginate scaffolds as nanocomposite scaffolds in this study.

The rationale of the study was to synthesize and characterize a novel biocompatible nanocomposite material for bone scaffold applications using the freeze-drying process. For the first time, this work reports the impact of incorporating Zn-Sr-BGNPs on the alginate scaffold properties. This study hypothesized that doping an alginate scaffold with Zn-Sr-BGNPs would have a strong beneficial effect on the regulation of cellular responses. The challenges of incorporating Zn-Sr-BGNPs in the alginate scaffold were addressed. The ratio of Zn in xZn-Sr-BGNPs (where x = 0. 1.0, and 1.5) and the ratio of Zn-Sr-BGNP: alginate in composite scaffold should directly affect bone formation. The presence of Zn-Sr-BGNP fillers in the composite scaffold had a beneficial effect on promoting bioactivity. The Zn-Sr-BGNPs were produced by a two-step post-functionalization according to a previous protocol [16]. The properties of the synthesized composite scaffolds with Zn-Sr-BGNP: alginate ratios of 2:1 and 1:1 were extensively characterized in terms of morphology, microstructure, apatite formation ability, swelling behavior, cytotoxicity, calcium formation, and cell adhesion.

## 2. Results and Discussion

### 2.1. Zn-Sr-BGNPs Characterization

SiO_2_-NPs were successfully prepared through the sol-gel process using ammonium hydroxide, the base catalyst, to control the size of the particles prior to the two-step post-functionalization process to obtain Zn-Sr-BGNPs. The uniform and monodispersed spherical Zn-Sr-BGNPs had a diameter range of 120 ± 20 nm, as shown in Figure 1. The nanoparticles exhibited smooth surfaces and regular shapes. The X-ray diffraction (XRD) patterns showed a broad halo peak at ~23° (2θ), which was assigned to the amorphous structure of Zn-Sr-BGNPs that contributes to the ion release capacity (right panel, Figure 1). This is consistent with what has been found in a previous report that the amorphous structure could facilitate biodegradation and enhance bioactivity [23]. Thus, Zn-Sr-BGNPs have great potential to induce bioactivity in composite scaffolds in this study through the released ions.

The quantitative analysis (wt%) of the Zn-Sr-BGNPs was measured using X-ray fluorescence (XRF). The XRF showed the presence of Si, Ca, Sr, and Zn in BGNPs (Table 1). The nominal ratios of 1.0:0.5:1.5:x SiO_2_:CaO:SrO:ZnO (where x = 0, 1.0, and 1.5) were used in this study. However, not all of the added Ca, Sr, and Zn was incorporated into the silica network. The silica network loosened as silica content reduced from 45.20% ± 1.09% to 36.03% ± 1.20%, suggesting the incorporation of network modifiers, including Ca, Sr, and Zn. The substitution of St for Ca expanded the glass network because the ionic radius of Sr is larger than that of Ca. Zn also played a role as a network modifier or intermediary. In addition, the total amount of Ca and Sr (network modifiers) was reduced when Zn was incorporated in the second post-functionalization step due to the high binding affinity of Zn with the glass network compared to that of the alkaline earth elements [24]. These results go beyond previous reports, showing that not all of the nominal network modifier cation was incorporated into the particles [25]. The transmission electron microscopy (TEM), XRD, and XRF results indicated the incorporation of Ca, Sr, and Zn in the amorphous network structure without the presence of excess salt on the surface of the particles. Incorporating Ca, Sr, and Zn did not affect the particle size, shape, and morphology. Our previous study reported the sustained release of Si, Ca, Sr, and Zn from BGNPs in a simulated body fluid (SBF) and phosphate-buffered saline (PBS) over 21 days [16]. Therefore, Zn-Sr-BGNPs were incorporated into the alginate scaffold to improve the bioactivity of the composite scaffolds in this study.

### 2.2. Zn-Sr-BGNP: Alg Scaffold Characterization

The porosity and surface structure of the scaffolds played an important role in cell attachment, adhesion, and three-dimensional cell growth. The pure alginate scaffolds and Zn-Sr-BGNP: Alg composite scaffolds were prepared using freeze-drying. The morphological structure of scaffolds was evaluated using scanning electron microscopy (SEM) (Figure 2a–g). The pore size was analyzed using ImageJ (Figure 2h). The SEM images indicated the highly interconnected porous structure of the composite scaffolds. The appropriate interconnection pores impacted on cell distribution, cell migration, cell differentiation and capillary formation [26,27]. In addition, the porous scaffold degradation immersed in an aqueous environment causes changes in the structure. Zn-Sr-BGNPs were dispersed in the alginate matrix. The surface morphology of the pure alginate scaffolds was smooth, while the rough surface of the composite scaffold was observed when different amounts of Zn-Sr-BGNPs were added. The average pore sizes of the pure alginate scaffold and the composite scaffolds were 123 ± 8.5 µm; and 70 ± 2.0, 95 ± 3.4, and 90 ± 0.0 µm for the 1:1 ratio group (0Zn/Alg, 1Zn/Alg, 1.5Zn/Alg); and 30 ± 1.2, 52 ± 2.7, and 51 ± 2.9 µm for the 2:1 ratio group (0Zn/Alg, 1Zn/Alg, 1.5Zn/Alg), indicating the significant reduction of pore size within the 2:1 ratio group. The pore size statistically significantly decreased (* *p* < 0.05) with increasing particle content, implying that the incorporated Zn-Sr-BGNPs compressed the pore size of the scaffold. The reduction in pore size was attributed to the fact that Zn-Sr-BGNPs could fill in the alginate crosslink chains due to their small size; the pores became interconnected with denser and thicker pore walls filled with increasing amounts of Zn-Sr-BGNPs [28]. Scaffold permeability was an important factor for nutrient and oxygen transportation. The porosity, interconnectivity, tortuosity, pore size, and shape of scaffolds directly affected scaffold permeability, degradation, and cell migration [29]. The surface area and pore volume of the composite scaffolds ranged from 11.57 to 30.64 m^2^·g^−1^ and 0.012 to 0.051 cm^3^·g^−1^, respectively (Table 2). An increase in BET surface area was observed in the 0Zn/Alg (1:1 and 2:1) composite scaffold because Sr-substituted BGNPs had a higher surface area and pore volume than Zn- and Sr-substituted BGNPs, implying a looser network structure. A reduction of surface area and pore volume was observed in the 2:1 ratio group (0Zn/Alg, 1Zn/Alg, 1.5Zn/Alg). The higher the amount of Zn-Sr-BGNPs added to the scaffold, the thicker the wall of the scaffold was. The texture analysis results confirmed that the incorporation of Zn-Sr-BGNPs caused the scaffold walls to thicken, reducing the surface area and pore volume of the scaffold. These basic findings are consistent with previous research showing that the incorporation of BGNPs in alginate scaffolds decreased the pore size of the composite scaffolds [28].

The attenuated total reflectance–Fourier transform infrared (ATR-FTIR) spectra (Figure 3) of the 0Zn/Alg (1:1), 0Zn/Alg (2:1), 1Zn/Alg (1:1), 1Zn/Alg (2:1), 1.5Zn/Alg (1:1), and 1.5Zn/Alg scaffolds showed the typical asymmetric stretching vibration peak of Si-O-Si in Zn-Sr-BGNPs at ~1100 cm^−1^ and the symmetric stretching vibration of the Si–O bond at ~800 cm^−1^, indicating the presence of Zn-Sr-BGNPs in the scaffolds. The characteristic band of asymmetric carboxyl at ~1625 cm^−1^ and broad hydroxyl band of alginate at ~3450 cm^−1^ of Alg were shifted for the Zn-Sr-BGNP: Alg composite scaffolds [23,30]. The XRD patterns of the scaffolds showed no diffraction peaks, except a broad band between 20 and 25° (2θ), which implies the amorphous nature of both alginate and Zn-Sr-BGNPs (Figure 4). The amorphous structure suggests an internal disorder that could increase bioactivity and the rate of biodegradation [31]. The amount of Zn-Sr-BGNPs in the composite scaffolds did not alter the amorphous nature, showing that the presence of Zn-Sr-BGNPs in the scaffolds did not change their degradation behavior. It is important to highlight the fact that these Zn-Sr-BGNP: Alg composite scaffolds can be degraded because of their amorphous nature. The results lead to the similar conclusion that the dissolution rate of amorphous phases was high compared to the crystalline phase [32].

### 2.3. Compression Testing of Scaffolds

The mechanical performance of pure Alg scaffolds and Zn-Sr-BGNP: Alg scaffolds was evaluated by compression testing. The compressive stress was calculated from maximum force and initial cross-sectional surface area (Figure 5a). The compressive stress of pure Alg scaffolds was 0.49 ± 0.08 MPa, 1:1 ratio group of 0Zn/Alg, 1Zn/Alg, and 1.5Zn/Alg were 2.00 ± 0.46, 4.73 ± 0.48, and 1.38 ± 0.33 MPa and 2:1 ratio group of 0Zn/Alg, 1Zn/Alg, and 1.5Zn/Alg were 0.78 ± 0.10, 1.58 ± 0.17, and 0.69 ± 0.06 MPa, respectively. These results showed that the incorporation of Zn-Sr-BGNPs in composite scaffolds (1:1 ratio) have a statistically significant effect on compressive stress (*p* = 0.05). A similar pattern of results was obtained when doping bioactive glass with different amounts of Zn in the composite scaffolds [19,23,33]. The pore size of composite scaffolds decreased upon incorporation of Zn-Sr-BGNPs, resulting in an increase in mechanical properties [34]. However, the compressive stress statistically significant decreased with increased amounts of Zn-Sr-BGNPs in composite scaffolds from 1:1 to 2:1 ratio groups. The reduction in mechanical performance of 2:1 ratio group was attributed to the fact that high amount of Zn-Sr-BGNPs could collapse the alginate crosslink network and inhibit the movement the alginate crosslink chains. In addition, the weak attractive forces between organic and inorganic directly affected the mechanical properties of the scaffold [35]. Together, the present findings indicated that the incorporation of appropriated amount of Zn-Sr-BGNPs in composite scaffolds (1Zn/Alg (1:1)) could dramatically improve the mechanical behaviors.

### 2.4. In Vitro Swelling

The swelling ability, a crucial parameter, directly affects the efficiency of nutrient penetration and metabolic waste transport within the scaffolds. A high swelling ratio in the fabricated scaffolds could maintain biological activity under physiological conditions. Thus, it is necessary to understand the swelling behavior. The swelling ratio of the pure alginate scaffold was higher than that of all the Zn-Sr-BGNP: Alg composite scaffolds, as shown in Figure 5b. The swelling ratio increased in a time-dependent manner. There was no significant change in the swelling ratio at 1 and 4 h incubation. However, incorporating 1Zn-Sr-BGNPs and 1.5Zn-Sr-BGNPs in the composite scaffolds significantly decreased the swelling ratio after 24 h of incubation. This might be because Zn has a high binding ability with the glass network, resulting in a more compact network in the dense BGNPs. Meanwhile, incorporating 0Zn-Sr-BGNPs (particles substituted with only Sr) in the composite scaffolds did not alter the swelling ratio compared to the pure alginate scaffold. This might be because doping 0Zn-Sr-BGNPs in the composite scaffolds slightly increased the surface area of the scaffold. The initial swelling ratio decreased from 0.32 to 0.19 (1 h), and the final swelling ratio decreased from 2.19 to 1.32 (14 d) in parallel with an increasing ratio of Zn-Sr-BGNP: Alg. Zn-Sr-BGNPs have a low ability to absorb fluid compared with highly hydrophilic alginate. In addition, the Zn-Sr-BGNPs incorporated in composite scaffolds increased the pore wall thickness, leading to a slow absorption rate. Zn-Sr-BGNPs acted as the filler and made a compact network. A decrease in swelling ration might be attributed to the small pore size of composite scaffolds compared to pure alginate [19].

### 2.5. In Vitro Bioactivity Assessment

The in vitro biomineralization of scaffolds was investigated by immersing them in the simulated body fluid (SBF) solution for 21 days and imaging them using SEM and EDX-SEM (Figure 6). The surface of the alginate scaffolds was smoother than that of other samples. An apatite-rich layer (cauliflower-like apatite) on the surface of the scaffolds formed more substantially on the surface of the Zn-Sr-BGNP: Alg composite scaffolds after soaking in the SBF solution for 21 days than on the alginate scaffolds. The formation of this apatite layer helped recruit cells to the scaffold and assisted with the cell attachment [28,36]. EDX analysis was conducted to determine the chemical composition of the mineral seen in the SEM images. The EDX-SEM results showed Ca and P on the alginate scaffold. Strong P signals appeared on the Zn-Sr-BGNP: Alg composite scaffolds. The Sr and Zn remained inside the scaffold, suggesting that the Zn-Sr-BGNPs persisted in the scaffold after 21 days of incubation (Figure 6b–g). The apatite formed mainly because of the ion exchange from the BG in the composite scaffolds [37]. Thus, incorporating Zn-Sr-BGNPs increased bioactivity by accelerating apatite formation through ion exchange with Zn-Sr-BGNPs in the composite scaffolds.

### 2.6. Cell Viability Study

The in vitro cell viability of treated MC3T3-E1 cells was investigated using the MTT assay compared to the control (untreated MC3T3-E1 cells). Figure 7 presents the cell viability after culturing for 24 and 72 h. The composite scaffolds exhibited no cytotoxic effects on the cells, indicating good biocompatibility. The relative cell viability was statistically significantly increased only in 1Zn/Alg (1:1) after culturing for 24 h (* *p* < 0.05), as shown in Figure 7a. The cell viability of the alginate scaffold and the Zn-Sr-BGNP: Alg composite scaffolds increased when culturing time increased from 24 to 72 h. The results showed a statistically significant difference between treated and untreated MC3T3-E1 cells after culture for 72 h (* *p* < 0.05), as shown in Figure 7b. Remarkably, the viability of cells treated with the 1:1 ratio group (0Zn/Alg, 1Zn/Alg, 1.5Zn/Alg) was higher than those treated with the 2:1 ratio group and pure alginate. Thus, the 1:1 ratio group (0Zn/Alg, 1Zn/Alg, 1.5Zn/Alg) composite scaffolds have great potential to stimulate cell growth.

### 2.7. Cell Attachment

The adhesion of cells to scaffolds is an important step associated with the proliferation, migration, and differentiation needed for successful tissue regeneration. The scaffolds functioned as the initial 3D temporary structure for cells based on their physical, chemical, and biological properties. The porosity, pore size, and interconnectivity of the scaffolds directly affected nutrient transport and cell growth [38]. The cell attachment and morphology on the scaffolds were assessed using SEM images. Cell attachment and cell spreading were detected, as shown in Figure 8. In addition, SEM imaging showed cells connecting with other cells, especially in the 1Zn/Alg (1:1) scaffold. These results confirmed cell-to-matrix and cell-to-cell interactions on the scaffolds, indicating that the scaffolds provided a suitable environment, supported the cell attachment, and maintained cell proliferation. Thus, incorporating 1Zn-Sr-BGNPs into the alginate at a 1:1 ratio could enhance the biological performance of the composite scaffold.

### 2.8. Calcium Deposition

Calcium mineralization is considered a marker for osteoblast cell differentiation, a critical step in bone regeneration [39]. Alizarin red staining is the gold standard for qualitatively identifying calcium accumulation. When cells were treated with scaffolds, calcium deposition increased with increasing culturing time from 7 to 14 days in all scaffold groups (Figure 9). There was a trend toward increased calcium deposition on Zn-Sr-BGNP: Alg composite scaffolds compared to pure alginate scaffolds. It has been reported that incorporating the BGs with alginate could improve the mechanical properties and the biological activity of the composite scaffolds [20,40,41]. Zn-Sr-BGNPs can influence the formation of mature osteoblasts in basal medium (with no osteogenic supplements). The findings of our study are consistent with our previous study showing that Zn-containing dense bioactive glass nanoparticles have the potential to stimulate osteogenic differentiation on PDLSCs [16]. Together, the initial findings indicated that the incorporation of Zn-Sr-BGNPs in Alg scaffolds should be considered potential candidates for artificial bone scaffolds. The key to success in bone tissue engineering consisted of the scaffold’s properties and cellular responses. The success or failure of bone regeneration is also based on inflammation and angiogenesis [42]. The lack of information on inflammation and angiogenesis is a boundary in this study. Thus, these issues might be addressed in future studies.

## 3. Materials and Methods

All reagents were from Sigma–Aldrich (Bangkok, Thailand) unless stated otherwise. Ethyl alcohol (99.5%), ammonium hydroxide, tetraethyl orthosilicate (TEOS), calcium nitrate tetrahydrate (99%), strontium nitrate (99%), zinc nitrate hexahydrate (≥98%), phosphate buffered saline (PBS), alginate, sodium chloride (NaCl), sodium hydrogen carbonate (Na-HCO_3_), potassium chloride (KCl), di-potassium hydrogen phosphate trihydrate (K_2_HPO_4_·3H_2_O), magnesium chloride hexahydrate (MgCl_2_·6H_2_O), hydrochloric acid (HCl), calcium chloride (CaCl_2_), sodium sulfate (Na_2_SO_4_), nitric acid, minimum essential medium eagle alpha (α-MEM, Gibco^TM^, Bangkok, Thailand), fetal bovine serum (FBS, Thermo Fisher Scientific, Bangkok, Thailand), Antibiotic-Antimycotic (Thermo Fisher Scientific), trypsin-EDTA (Thermo Fisher Scientific), sodium bicarbonate (NaHCO_3_), 3-(4,5-dimethylthiazol-2-yl)-2,5-diphenyltetrazolium bromide (MTT, Thermo Fisher Scientific), dimethyl sulfoxide (DMSO), paraformaldehyde, Alizarin Red S.

### 3.1. Zn-Sr-BGNPs Synthesis and Characterization

Zn-Sr-BGNPs were Zn-Sr-BGNPs using the sol-gel process and two-step post functionalization described in previous work [16]. Silica nanoparticles (SiO_2_-NPs) with diameter size range 120 ± 20 nm were first synthesized prior to post-functionalization to incorporate Ca, Sr, and Zn. Briefly, 74.1 mL of Milli-Q water, 592.5 mL of ethanol (99.5%), and 8.7 mL of ammonium hydroxide were mixed in a 1 L Erlenmeyer flask at a stirring rate of 700 rpm for 15 min. Then, 45.0 mL of tetraethyl orthosilicate (TEOS) was added into the prepared solution and left on the stirrer for at least 8 h to complete the reactions. After that SiO_2_-NPs were collected by centrifugation at 5000 rpm for 30 min and then were simultaneously washed with ethanol (two times) and distilled water (one time) to remove unreacted substances. The obtained SiO_2_-NPs were suspended in Milli-Q water.

Calcium nitrate tetrahydrate and strontium nitrate, used as the precursors of Ca^2+^ and Sr^2+^, were added to the SiO_2_-NP suspension with a nominal ratio of 1.0 SiO_2_: 0.5 CaO: 1.5 SrO and left in the ultra-sonication bath. After 1 h, the white pellets of particles were collected by centrifugation at 5000 rpm for 20 min. The white pellets were then dried at 60 °C overnight followed by calcination at 680 °C for 3 h at a heating rate of 3 °C/min to obtain 0Zn-Sr-BGNPs. Zinc nitrate hexahydrate was added to 0Zn-Sr-BGNPs with a nominal ratio of 1.0 SiO_2_:0.5 CaO:1.5 SrO:x ZnO (where x = 1.0 and 1.5). These particles were calcined at 550 °C for 3 h at a heating rate of 3 °C/min to form 1Zn-Sr-BGNPs and 1.5Zn-Sr-BGNPs. The particles were then washed with ethanol twice as shown in Figure 1. To investigate the surface morphology and particle size of Zn-Sr-BGNPs, a Trans-mission Electron Microscope (TEM, JEOL 1400 operated at 120 kV) was used. The dried particles were suspended in ethanol and collected on TEM grids. To detect the crystalline structure, XRD (Bruker AXS/D8Discover) with a Bruker AXS automated powder diffractometer using Cu Kα radiation (1.540600 A°) at 40 KV/40 mA was used. Data was collected in the 10–70° 2θ range with a step size of 0.02° and a dwell time of 1.0 s. To determine the elemental composition of Zn-Sr-BGNPs, X-ray fluorescence (XRF: Fisfer/XUV773) with X-ray generators in at 20 kV operating in a vacuum was used and measuring time was 1 min.

### 3.2. Zn-Sr-BGNP: Alg Scaffold Fabrication and Characterization

Figure 2 represented the Zn-Sr-BGNP: Alg scaffold preparation. Zn-Sr-BGNP: Alg porous scaffold was prepared by mixing sodium alginate and Zn-Sr-BGNPs in different ratios as shown in Table 3. Zn-Sr-BGNPs and alginate were weighted and suspended in 50 mL DI water at a stirring rate of 500 rpm for 6 h at room temperature to form a gel-like texture. Then, samples were frozen and lyophilized for 24 h in a freeze drier at −50 °C (Lyovapor L-200, Buchi). Calcium chloride (CaCl_2_) was used to form a network connection. To form the crosslink, dried samples were soaked in 5%(*w*/*v*) CaCl_2_ for 20 min. After that, the soaked solution was removed and washed with DI water. The samples were then lyophilized for 24 h to obtain Zn-Sr-BGNP: Alg porous composite scaffolds.

To investigate the morphology and pore size of the Zn-Sr-BGNP: Alg scaffold, a scanning electron microscope (SEM, JEOL, JSM-6610 LV) was used. The sample was coated with gold (Sputter Coater, Cressington, 108 Auto). The condition of the high voltage was 5.00 kV, the spot size was 3.0, and the detector was EDT. To analyze the pore size, the ImageJ program was used. To identify the compound type and crystal structure of compounds, X-ray Diffractometer (XRD, Bruker AXS/D8Discover)) was used. XRD pattern was collected with a Bruker AXS automated powder diffractometer using Cu Kα radiation (1.540600 A°) at 40 KV/40 mA. Data was collected in the 15–80° 2θ range with a step size of 0.02° and a dwell time of 1.0 s. To analyze changes in the chemical structure of scaffolds, Fourier transform infrared spectroscopy (FTIR; Thermo Scientific Nicolet iS5) was used in attenuated total reflection (ATR) mode at the wavenumber ranging from 4000 to 400 cm^−1^ at the scan speed 32 scan/min with a resolution of 4 cm^−1^. To determine the surface area and pore volume of scaffolds, a surface area and porosity meter (Specific Surface Area Analyzer (BET), BELSORP-mini II, BEL) was used.

### 3.3. Compression Testing of Scaffolds

To investigate the mechanical property, mechanical tests were carried out on scaffolds under compression mode using Shimadzu AGS-X 500N. The scaffolds were cut into cylinders of 10 mm diameter and 10 mm height (Φ 10 mm × h 10 mm) with plane-parallel ends. The height and diameter were measured using a digital vernier caliper. A crosshead speed was set at 2 mm·min^−1^ using a load cell of 500 N at room temperature. Each sample was compressed to approximately 50% of its initial height. The compressive stress of pure Alg scaffolds and Zn-Sr-BGNP: Alg scaffolds were calculated from maximum force and initial cross-sectional surface area.

### 3.4. In Vitro Swelling

The water uptake ability of scaffolds was measured by soaking them in phosphate-buffered solution (PBS) for 1, 4, 8, 24, 168, and 336 h. After eliminating the surplus water with surface wiping, the scaffolds were removed and weighed while wet. Three specimens of each composition were measured, and the average value was used to calculate the water uptake ability. To calculate water absorption, the following Equation (1) was used.
(1)$$Swelling\;ratio=\frac{W_{2}-W_{1}}{W_{1}}$$
where W_1_ is the dry weight of the samples and W_2_ is the weight of the samples after immersion.

### 3.5. In Vitro Bioactivity Assessment

The in vitro bioactivity assessment was conducted by incubating scaffolds in the SBF solution at pH 7.4 37 °C shaking at 120 rpm for 21 days. At the end of the incubation period, the scaffolds were collected. The morphological structures and composition of Zn-Sr-BGNP: Alg scaffolds were detected using SEM (JEOL, JSM-6610 LV) and EDX-SEM (OXFORD, INCAx-act), respectively. The SBF solution was prepared following a previous study [43].

### 3.6. Cell Viability Study

#### 3.6.1. Cell Culture

MC3T3-E1 cells (ATCC^®^ CRL-2593^TM^) of the murine pre-osteoblast cell line, were routinely cultured in T-75 flask under standard condition in a humidified atmosphere at 37 °C and 5% CO_2_ in the α-MEM (Thermo Fisher Scientific), supplemented with 10% fetal bovine serum (FBS, Thermo Fisher Scientific) (*v*/*v*), 100 U/mL Antibiotic–Antimycotic (Thermo Fisher Scientific). Cells were passaged by trypsinising using trypsin-EDTA (500 μg/mL) (Thermo Fisher Scientific) upon confluence (70–80%) and re-suspended in the α-MEM before cells were counted. The cell stock was diluted to the desired concentration (5 × 10^4^ cells/mL).

#### 3.6.2. Cell Viability

To evaluate the cytotoxicity effect of scaffolds on cell viability, MTT colorimetric assay (Thermo Fisher Scientific) was used according to the manufacturer’s instructions. MC3T3-E1 cells were seeded in the flat-bottomed 96-well plates (Corning) with the cell concentration at 5 × 10^3^ cells/well. The cells were incubated at 37 °C overnight to allow the cell attachment. The media was replaced with extracted media (indirect method). To prepare extracted media, 75 mg of dried scaffolds were incubated in 50 mL of plain α-MEM in an incubator shaker at 37 °C 120 rpm for 24 h and then were filtrated through a sterilized filter. After scaffolds were removed from the incubated media, 10% FBS (*v*/*v*) and 100 U/mL Antibiotic–Antimycotic were supplemented. Cells were exposed to dissolution media for 24 and 72 h. Cell viability was determined using the MTT colorimetric assay based on the conversion of 3-(4,5-dimethylthiazol-2-yl)-2,5-diphenyltetrazolium bromide (MTT) into formazan. The formazan is soluble in dimethyl sulfoxide (DMSO) and the concentration of soluble formazan was determined using a microplate reader (Infinite^®^ 200 Tecan, Austria) at 570 nm. The relative cell viability (% viability compared to untreated cells: control) was calculated as mean value ± standard error of the mean (n = 6). A reduction in cell viability by more than 30% is considered a cytotoxic effect (cell viability less than 70%: ISO 10993-5).

### 3.7. Cell Attachment and Morphology Study

SEM was used to investigate the attachment and morphology of the cells seeded on the scaffolds, operated at 15 kV. After 24 and 72 h of culturing on the scaffolds, the cells were fixed with 4% paraformaldehyde for 1 h at room temperature. Then, the cells were dehydrated by immersing in a gradient of ethanol solutions of 30%, 50%, 70%, 90%, and 100%. The attached cells were imaged using SEM.

### 3.8. Calcium Deposition

To evaluate the mineralization nodules in vitro of MC3T3-E1 cells, alizarin red S staining was used. MC3T3-E1 cells were seeded and cultured in the scaffolds with a cell concentration of 1 × 10^4^ cells/mL. Cell culture media were changed twice a week for 14 days. After 7 and 14 days in culture, the cells were fixed with 4% paraformaldehyde for 30 min. Then, the fixed cells were stained with 2% alizarin red S in PBS at pH 4.2 to detect calcified tissue formation. Images were obtained with an inverted optical microscope (LABOMED TCM400) using ToupView program.

### 3.9. Statistical Analyses

The graphs shown present the results as the mean value with the standard deviation (SD) as the error bars. All the quantitative experiments were carried out at least in triplicate. Statistical analyses were performed by one-way analysis of variance (ANOVA) in Minitab with the appropriate post hoc comparison test (Tukey’s test). A *p*-value < 0.05 was considered significant.

## 4. Conclusions

Zn-Sr-BGNPs with a diameter range of 120 ± 20 nm were synthesized through the sol-gel and two-step post-functionalization processes. Sr and Zn were successfully substituted for Ca in the BGNPs while maintaining size, morphology, and amorphous structure. The ratio of zinc in Zn-Sr-BGNPs and ratio of Zn-Sr-BGNPs in composite porous scaffolds (Zn-Sr-BGNPs: Alg) were characterized. The composite scaffolds had physiochemical and biological properties crucial to inducing bone formation. The composite scaffolds had an appropriate pore size that ranged from 50 to 120 μm. Incorporating Zn-Sr-BGNPs did not significantly alter the surface area, pore volume, or amorphous structure. Incorporating Zn-Sr-BGNPs improved the mechanical properties of the scaffolds. The swelling ratio decreased due to the incorporation of Zn-Sr-BGNPs leading to increased pore wall thickness. Incorporating Zn-Sr-BGNPs induced apatite formation due to the fact that ion exchange from Zn-Sr-BGNPs in the composite scaffolds did not alter the cell viability of MC3T3-E1, facilitated cell attachment, and enhanced calcium deposition. Thus, Zn-Sr-BGNP: Alg has the great potential to be used as a bone scaffold in bone tissue engineering. However, further investigation of behaviors, interactions, and responses of cells and these scaffolds, such as inflammation, angiogenesis, and osteogenesis, will be needed.

## Data Availability

The datasets generated and/or analyzed during the current study are available from the corresponding author on reasonable request.

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
