# Peer review of "Zinc and Strontium-Substituted Bioactive Glass Nanoparticle/Alginate Composites Scaffold for Bone Regeneration"

_ijms, 2023, doi:10.3390/ijms24076150_

Round 1

Reviewer 1 Report

1.      The abstract requires the addition of quantitative results.

2.      Please end your abstract with a "take-home" message.

3.      Describe the novelty of the article made by the author? From the results of my evaluation, it seems that many similar published works adequately explain what you have raised in the current manuscript. If there is something others really new in this manuscript, please highlight it more clearly in the introduction section.

4.      The work, novelty, and limitations of similar prior studies must be explained in the introduction section to highlight the research gaps that the current study aims to fill.

5.      The authors need to explain the basic concept of degradation in scaffold for giving better understand to the reader. Suggested reference needs to incorporated as follows: The Effect of Tortuosity on Permeability of Porous Scaffold. Biomedicines 2023, 11, 427. https://doi.org/10.3390/biomedicines11020427

6.      In the whole of the manuscript, the authors sometimes made a paragraph only consisting of one or two sentences that made the explanation not clearly understood. The authors need to extend their explanation to become a more comprehensive paragraph. In one paragraph, it is recommended to consist of at least 3 sentences with 1 sentence as the main sentence and the other sentences as supporting sentences.

7.      In line 367-368, what is the basis of statistical analysis? Please giving more explanation about it.

8.      In the materials and methods, the authors need to add additional illustrations as a form of figure that explains the workflow of the present study to make the reader easier to understand rather than only the dominant text as a present form.

9.      More information about tools, such as the producer, country, and specifications, should be included.

Author Response

Dear Reviewer,

Thank you for your advice. The authors provided a point-by-point response.

Best regards,

Parichart Naruphontjirakul, PhD.

Reviewer 2 Report

In this manuscript the authors synthesized a novel biocompatible nano-composite material for bone scaffold applications through freeze-drying process by incorporating Zn-Sr-Bioactive glass nanoparticles (BGNPs) in an alginate scaffold. The scaffolds were further characterized in terms of morphology, microstructure, apatite formation ability, swelling behaviour, cytotoxicity, calcium formation and cell adhesion. The article is well written but in my opinion some characterizations are missing in the experimental part. In fact the author did not evaluate the mechanical properties of the composite scaffold highlighting the effect of the nanoparticles on the mechanical properties of the Alg scaffolds. Moreover , as reported also by the authors, the BGNPS as well as the Zn can modify the antibacterial activity and the anti-cancer properties of Alg scaffold, but the release of Zn and BGNPs have not been evaluated by the authors in the present work. This is why that I would suggest the acceptance of the present manuscript in International Journal of Molecular Sciences after some major revisions that I report here:

-) In table 2 the authors reported the texture analysis of the composite scaffold. however if the pore volume seems to decrease by increasing the nanoparticle amount the surface are has a complete different trend. Could the author better explain this behaviour.

-) In the label of figure 1 three different composition have been reported “0Zn-Sr-BGNPs, 1Zn-Sr-BGNPs, and 2Zn-Sr-BGNPs” however in the manuscript the 2Zn-Sr-BGNPs have  been never reported. Please correct the label or explain the presence od a further scaffold composition.

-) Figure 2 reports the morphological characterization of the Alg and Alg composite scaffolds with the average pore size. however the authors only report the results of the porosity analysis without any explanation of the pore size difference between the different scaffold compositions. Please give na explanation of the different morphology found in the scaffold proposed in this manuscript.

-) Mechanical properties of the scaffolds have not been evaluated. In particular the effect of the BGNPs on the scaffold properties have not been reported. Please report the mechanical properties of the ALg and Alg composite scaffold.

-) As the authors stated the presence of the BGNPS as well as the Zn can modify the antibacterial activity and the anti-cancer properties of Alg scaffold, but they did not evaluate the release profile of this particle from the scaffolds. Please report the release properties of Zn as well as of BGNPs from Alg scaffold in physiological environment.

-) Pp 7 line 176 please correct “bine” with “bind”.

-) In chapter 3.5.2 the author describes the method to perform the cell viability test. I have only one concern about the method used. In fact, to prepare the extracted media the authors filtrated ad sterilized it before use. However, if the scaffolds release some particles, it would be retained by the filter and so they did not evaluate the real effect of scaffold release on cell behaviour. I believe that the experiment must be done again without filtering the extracted media before the contact with cell.

-) To evaluate the mineralization capability of composite scaffold onto MC3T3-E1 scaffold the authors used alizarin red staining. However, to further evaluate the osteoinduction capability of the scaffold I would suggest performing an ALP assay after 7, 14 and 21 days.

Author Response

(The authors gave the same response as above.)

Reviewer 3 Report

In the submitted manuscript, the authors have formulated zinc and strontium-substituted bioactive glass nanoparticle (Zn-Sr-BGNP)/alginate composite scaffolds and showed its possible application for bone regeneration. Alginate is well-known for its high biocompatibility. However, when contacted with a physiological environment, it turns into softer hydrogel. Thus, Zn and Sr containing BGNPs were used to modify the properties of alginate scaffolds in this study. Additionally, BGNPs provide high porosity, control particle size uniformity, increase purity and enhance bioactivity. After using different Zn to Si nominal ratios and ratios of Zn-Sr-BGNPs in alginate scaffold, the authors concluded that 1Zn-Sr-BGNPs: Alg (1:1) composite scaffold promoted bioactivity compared to the other composite scaffolds used in the study.  

Overall, the manuscript is nicely written. The conclusions are substantiated by the data. Similarly, the discussion touches on related work in the field and highlights the importance of the findings. However, there are some areas where further changes might help to improve the quality of this manuscript. 

·       Abstract: 

1.     Line 26: I am not sure if authors are willing to say ‘though’ or through.

·       Introduction: 

1.     Line 35: The ability in self-remodeling of bone is limited in replacing…tissue transplant. Kindly rewrite this sentence.

2.     Line 40: a synthesis bone graft…please check the used words.

3.     Line 50: because of lack mechanical strength…kindly correct it as because of lack of mechanical strength.

4.     Line 68: The sol-gel technique provides…bioactivity of BGNPs. Rewrite this sentence.

·       Results and Discussions: 

1.     Missing subtitles for the results and discussion section. Results should be described under representative subtitles.

2.     Line 176: Zn has the high ability to ‘bine’ with the glass network…use correct word to replace bine.

3.     Line 187: There is a missing full form for SBF solution.

4.     Line 207: The relative cell viability was statistically ‘significant’ increased…as shown in Figure 7(a). Please correct the sentence.

5.     Line 225: Kindly correct the word ingrowth. 

6.     The contribution or effect of incorporation of Sr in Zn-Sr-BGNPs is not discussed in this section.

7.     Limitations of the study need to be specified.

·       Figures: 

1.     Figure 1: Scale bar are not easily visible for 1 Zn-Sr-BGNPs and 1.5 Zn-Sr-BGNPs figures.

2.     Figure 2: Figures (a-g) need to be correctly ordered. Current sequence (a,b,d,f,c,e,g) is difficult to follow.

3.     Figure 6: Ca and P signals should be pointed out in the EDX-SEM figures.

4.     Figure 7: Authors should mention the p value for figure 7a and 7b for the respective groups. It is necessary to check how statistically different these groups are from each other. If it is not possible to mention the values in the figure, authors should provide those in the description.

5.     Figure 9: Calcium deposits are visualized through Alizarin Red S staining. It is interesting that in groups 1Zn/Alg (1:1), 1Zn/Alg (2:1) and 1.5Zn/Alg (1:1), staining is significantly increased after 14 days culture without osteogenic supplement. Can authors provide quantitative data for this experiment?   

·       Materials and Methods:

1.     Line 276-283: Kindly check the line spacing.

2.     Line 317: Full form of SBF is missing.

3.     Calcium deposition: Need to provide more details about the procedure and culture media containing scaffolds.

4.     Line 361: alizarin red s solution was stained with 2% alizarin red s in PBS…Kindly rewrite this sentence.

·       Conclusion:

1.     Line 382: Replace word grate with great.

Author Response

Dear Reviewer,
Thank you very much for giving us your wonderful advice. 
The authors provided a point-by-point response.
Please see the attachment.
Best regards,
Parichart Naruphontjirakul, PhD.

Reviewer 4 Report

-The manuscript in this form cannot be accepted, it needs revisions.

-There are conceptual and spelling errors, it must be re-read because there are often unconnected sentences. There are also typing errors.

-Amorphous state is often associated with bioactivity, this is not entirely true, because pure glass is amorphous but not bioactive, it is the composition of the glass and all the mechanisms of dissolution and precipitation that make it so. Consequently, you must rewrite the sentences

-Table 1 XRF can have two decimal places?

-Missing error in BET

-Why do you swell in PBS? and not SBF since bioactivity is then done in SBF?

-Figure 6 EDS is not clear, you cannot see element symbols, you also have to calculate the Ca/P ratio to say that the layer is apatitic. Also it would be good to do XRD to demonstrate HA formation and better compare the behaviour of different glasses.

Author Response

(The authors gave the same response as above.)

Round 2

Reviewer 1 Report

1.      In line 150 the authors explain about porosity. Please improve the explanation of porosity since it is impacting the performance of scaffold. The suggested relevant reference needs to adopted as follows: Level of Activity Changes Increases the Fatigue Life of the Porous Magnesium Scaffold, as Observed in Dynamic Immersion Tests, over Time. Sustainability 2023, 15, 823. https://doi.org/10.3390/su15010823

2.      Error and tolerance of experimental tools used in this work are important information that needs to be explained in the manuscript. It is would use as a valuable discussion due to different results in the further study by other researcher.

3.      Outcomes must be compared to similar past research.

4.      The authors need to improve the discussion in the present article become more comprehensive. The present form was insufficient.

5.      What are the limitations of the current work? Please include it before the concluding section.

6.      Mention further research in the conclusion section.

7.      The reference needs to be enriched from the literature published five years back. MDPI reference is strongly recommended.

8.      Suggested reducing the number of works from the authors in the reference.

9.      The authors were encouraged to proofread their work due to grammatical problems and linguistic style.

10.   Graphical abstract is encouraged to provide in submission after review.

Author Response

Dear Reviewer,

Thank you for your kind advice. The authors provided a point-by-point response to your comments

Sincerely yours,

Parichart Naruphontjirakul, PhD.

Reviewer 2 Report

The authors replied to all the reviewer comments improving the scientific soundness of the work. Some concerns remain about the mechanical properties f other scaffold. In m y opinion I would suggest performing the mechanical properties because thy are directly linked with the biological performance of the scaffolds. However, I did not fully agree with the statement “The porosity and swelling behaviour of scaffolds directly impacted on cell adhesion and the supply of nutrients. Swelling behaviour affects the mechanical property of the scaffolds. The pore size and swelling ratio of composite scaffolds decreased upon incorporation of BGNPs, resulting in an increase in mechanical properties [30]. Zn-Sr-BGNPs acted as the filler and made a compact network. A decrease in swelling ration might be attributed to the small pore size of composite scaffolds compared to pure alginate [18].”

In fact, a lower swelling degree could lead to a denser structure and higher mechanical properties but in this case the presence of nanoparticles could seriously affect the final mechanical properties of the composite scaffold more than the swellin properties. This is why I would suggest removing this sentence and analyse in deep the effect of the nanoparticles on the final mechanical properties of the scaffolds.

Author Response

Dear Reviewer,

Thank you for your kind advice. The authors have added the mechanical testing section.

Sincerely yours,

Parichart Naruphontjirakul, PhD.

Reviewer 4 Report

no comment

Author Response

Dear Reviewer,

Thank you for your kind response.

Sincerely your,

Parichart Naruphontjirakul, Ph.D.

Round 3

Reviewer 1 Report

I have minor revision that needs to addressed at this stage.

1.      Additional figure in introduction section is mandatory for giving better explanation in the present manuscript.

2.      In introduction section, please highlight the reason for performing in vitro (experimental) study? Why not in vivo (clinical), or in silico (computational) study? The methods have some advantages and disadvantages that needs to give an explanation. The authors encouraged to refer the relevant reference from Jamari et al. for this purpose: Adopted Walking Condition for Computational Simulation Approach on Bearing of Hip Joint Prosthesis: Review over the Past 30 Years. Heliyon 2022, 8, e12050. https://doi.org/10.1016/j.heliyon.2022.e12050

3.      Potential further study adopting In silico (computational) needs to explained in the discussion section since it bring several advantages compared to in vitro (performed study in the submitted work) and in vivo such as lower cost and faster results. In silico have been widely adopted in medical study for help faster the advance research, exclusively as preliminary before conducting in vitro or in vivo followed study in the further. See the suggested reference in comment number 1 from Jamari et al. and incorporated the used reference from this literature number 1, 11, 33, 50, 65, 72, and 83.

Author Response

Dear Reviewer,

Thank you for your comments.

The authors agreed that in vivo and in silico studies had their benefits as you had commented. But our work was performed in vitro as it is a general step to conducting research and obtaining basic information for future study. We already stated in our aims that doping an alginate scaffold with Zn-Sr-BGNPs would have a strong beneficial effect on regulating cellular responses. The challenges of incorporating Zn-Sr-BGNPs in the alginate scaffold were addressed. 

We agree that in vivo study is the critical step to success in tissue engineering. We plan to do this in future work. But in vivo and in silico studies are beyond the scope of this manuscript. We respect the value of research studies in different fields (in vitro, in vivo, and in silico). We think that adding the in silico in this manuscript is unnecessary.  

Best regards,